# Combined Effects of Race and Socioeconomic Status on Cancer Beliefs, Cognitions, and Emotions

**DOI:** 10.3390/healthcare7010017

**Published:** 2019-01-24

**Authors:** Shervin Assari, Pegah Khoshpouri, Hamid Chalian

**Affiliations:** 1Department of Psychiatry, University of Michigan, Ann Arbor, MI 48104, USA; 2Department of Psychology, University of California, Los Angeles (UCLA), Los Angeles, CA 90095, USA; 3Russell H. Morgan Department of Radiology and Radiological Sciences, Johns Hopkins University School of Medicine, Baltimore, MD 21205, USA; pkhoshp1@jhmi.edu; 4Department of Radiology, Duke University Medical Center, Durham, NC 27710, USA; hamid.chalian@duke.edu

**Keywords:** perceived risk, worries, cancer beliefs, cancer screening, Health Information National Trends Survey (HINTS), race, class, socioeconomic status

## Abstract

Aim: To determine whether socioeconomic status (SES; educational attainment and income) explains the racial gap in cancer beliefs, cognitions, and emotions in a national sample of American adults. Methods: For this cross-sectional study, data came from the Health Information National Trends Survey (HINTS) 2017, which included a nationally representative sample of American adults. The study enrolled 2277 adults who were either non-Hispanic Black (*n* = 409) or non-Hispanic White (*n* = 1868). Race, demographic factors (age and gender), SES (i.e., educational attainment and income), health access (insurance status, usual source of care), family history of cancer, fatalistic cancer beliefs, perceived risk of cancer, and cancer worries were measured. We ran structural equation models (SEMs) for data analysis. Results: Race and SES were associated with perceived risk of cancer, cancer worries, and fatalistic cancer beliefs, suggesting that non-Hispanic Blacks, low educational attainment and low income were associated with higher fatalistic cancer beliefs, lower perceived risk of cancer, and less cancer worries. Educational attainment and income only partially mediated the effects of race on cancer beliefs, emotions, and cognitions. Race was directly associated with fatalistic cancer beliefs, perceived risk of cancer, and cancer worries, net of SES. Conclusions: Racial gap in SES is not the only reason behind racial gap in cancer beliefs, cognitions, and emotions. Racial gap in cancer related beliefs, emotions, and cognitions is the result of race and SES rather than race or SES. Elimination of racial gap in socioeconomic status will not be enough for elimination of racial disparities in cancer beliefs, cognitions, and emotions in the United States.

## 1. Background

Race [1,2] as well as socioeconomic status (SES) [3,4,5,6,7,8,9,10] impact health outcomes. There is, however, a debate regarding whether the effects of race on health outcomes are fully due to lower SES of the minority groups or not [11,12,13,14,15,16,17,18,19]. Similarly, while race [20,21,22,23,24,25,26,27] and SES [28,29] both impact cancer incidence and outcomes, it is still unknown to what degree SES explains the racial gap in cancer outcomes [30,31,32,33,34,35,36]. While poor cancer beliefs are more common in racial minority groups as well as individuals with low SES [37,38,39,40], we still do not know whether all of the racial disparities in cancer beliefs are due to SES differences between races or race influences cancer beliefs above and beyond SES. A considerable amount of research suggests that SES only partially explains the racial differences in health [11,12,13,14,15,16,17,18,19], a finding which is also shown for cancer outcomes [24,25,30,31,32,33,34,35,36].

In other terms, it is still unknown whether it is race and SES or race or SES which shape cancer disparities [11,24,25]. If it is race or SES, then racial differences in cancer outcomes are fully explained by SES. In such case, eliminating SES gap across racial groups would be enough for elimination of racial gap in cancer outcomes. If it is race and SES, however, SES would only partially account for racial differences in outcomes [12,13,14,15,16,17,18,19]. In this case, elimination of racial gap in cancer outcomes would require interventions and programs that go beyond equaling SES across racial groups [41,42,43,44,45]. Residual effect of race that is above and beyond SES may be due to racism, discrimination, and culture [41,45]. Thus, understanding of whether SES fully mediates the effect of race on cancer outcomes has practical implications for public and social policy, public health programs, as well as clinical practice.

A considerable body of research has shown that low SES people and Black individuals have a higher risk of cancer [35,46,47,48], probably due to environmental exposures and behaviors such as poor diet, drinking, and smoking [49]. At the same time, Blacks and low SES people have lower health literacy [50] and a lower trust to the health care system as well as a lower perceived risk of cancer [51,52]. As a result, compared to high SES and White individuals, Black and low SES people have a lower tendency for cancer screening behaviors [53]. As a result, when diagnosed with cancer, cancer is at a more advanced stage, which reduces survival, and worsens the prognosis [35]. These all result in what we know as racial and SES disparities in cancer outcomes [35,46,47,48,54].

Aims: To expand the current knowledge on this topic, we used a national sample of American adults to test the separate and additive effects of race and SES on fatalistic cancer beliefs, perceived risk of cancer, and cancer worries. The implication of such knowledge will help with designing and implementing the most effective policies, programs, and practices that may eliminate the racial and SES gaps in cancer beliefs, cognitions, and emotions.

## 2. Methods

### 2.1. Design and Setting

Using data from the Health Information National Trends Survey (HINTS-5, Cycle 1, 2017), this was a cross-sectional study. HINTS is a national survey which has being periodically administered by the National Cancer Institute (NCI) since 2003. The HINTS study series provides a nationally representative picture of Americans’ cancer related information [55]. HINTS-5, Cycle 1 data were collected between January and May 2017 [56,57,58].

### 2.2. Ethics

All participants provided informed written consent. The Westat’s Institutional Review Board (IRB) approved the HINTS-5 study protocol (Westat’s Federal wide Assurance (FWA) number = FWA00005551, Westat’s IRB number = 00000695, the project OMB number = 0920-0589). The National Institute of Health (NIH) Office of Human Subjects exempted the HINT from IRB review.

### 2.3. Sampling

The HINTS sample is composed of American adults (age ≥ 18) who were living in the US and were not institutionalized. The HINTS-5, Cycle 1 used a two-stage sampling design in which the first stage was a stratified sample of residential addresses. Any non-vacant residential address was considered eligible. The address list was obtained from the Marketing Systems Group (MSG). In the second sampling stage, one adult was sampled from each selected household. The sampling frame composed of two strata based on concentration of minorities (areas with high and areas with low concentration of racial and ethnic minorities). Equal-probability sampling was applied to sample households from each stratum [55].

### 2.4. Surveys

The surveys were mailed to the participants’ addresses. A monetary incentive was given to the participants (included in the mails) to increase the participation rate. Two specific toll-free numbers were provided for the respondents to call: one number for English calls and one number for Spanish calls. The overall response rate was 32.4% [55].

### 2.5. Study Variables

The study variables included race, age, gender, educational attainment, income, history of cancer in family, health insurance status, cancer worries, fatalistic cancer beliefs, and perceived risk of cancer. Outcome measures included cancer beliefs, perceived risk of cancer, and cancer worries. Race/ethnicity was the independent variable. Educational attainment and income were mediators. Age, gender, history of cancer in family, and health insurance status were covariates.

#### 2.5.1. Independent Variable

**Race/ethnicity.** Race/ethnicity was the independent variable of interest. Race/ethnicity was treated as a dichotomous variable (0 non-Hispanic Whites, 1 non-Hispanic Blacks).

#### 2.5.2. Covariates

**Demographic Factors.** Age and gender were the demographic covariates. Age was an interval measure ranging from 18 to 101. Gender was treated as a dichotomous variable (0 female, 1 male).

**Health Insurance Status.** Availability of health insurance was measured using the following insurance types: (1) Insurance purchased from insurance companies; (2) Medicare (for people 65 and older, or people with disabilities); (3) Medicaid, Medical Assistance, or other government-assistance plans; (4) TRICARE and any other military health care; (5) Veterans Affairs; (6) Indian Health Services; and (7) any other health coverage plan. Insurance status was operationalized as a dichotomous variable (0 no insurance, 1 any insurance, regardless of its type).

**Family History of Cancer.** History of cancer in the family was asked using the following single item. “Have any of your family members ever had cancer?” The answers included yes, no, and do not know.

#### 2.5.3. Dependent Variables

**Fatalistic Cancer Beliefs.** Fatalistic cancer beliefs were measured using the stem “How much do you agree or disagree with each of the following statements?” followed by the following items: (1) There’s not much you can do to lower your chances of getting cancer”; (2) “It seems like everything causes cancer”; (3) “There are so many different recommendations about preventing cancer, it’s hard to know which ones to follow”; and (4) “When I think about cancer, I automatically think about death”. Answers included four-response Likert items ranging from strongly disagree to strongly agree. A sum score was calculated, with a possible range from four to sixteen. Fatalistic cancer beliefs were operationalized as an interval measure, with higher scores reflecting higher fatalistic beliefs [59].

**Perceived Risk of Cancer.** Perceived risk of cancer was measured using the following item: *“How likely are you to get cancer in your lifetime?”* Responses were on a five item Likert scale ranging from (1) very unlikely to (5) very likely. Perceived risk of cancer was operationalized as an interval measure, with a higher score indicative of higher perceived cancer risk [60].

**Cancer Worries.** Cancer worries were measured using the following item: *“How worried are you about getting cancer?”* Responses were on a 5-item response, items from (1) not at all to (5) extremely high. Cancer worries were operationalized as an interval measure, with a higher score indicating more cancer worries [61].

#### 2.5.4. Mediators

**Educational Attainment.** Educational attainment, one of the main SES indicators, was the mediator in this study. Educational attainment was treated as an interval variable ranging from 1 to 5: (1) less than high school graduate, (2) high-school graduate, (3) some college education, (4) completed bachelor’s degree, and (5) having post-baccalaureate degrees. Educational attainment ranged from 1 to 5, with a higher score indicating higher SES.

**Income.** Income, one of the most robust SES indicators, was the other mediator in this study. Income was treated as an interval variable ranging from 1 to 5: (1) Less than $20,000; (2) $20,000–34,999; (3) $35,000–49,999; (4) $50,000–74,999; (5) $75,000 or more. Income ranged from 1 to 5, with a higher score indicating higher SES.

### 2.6. Statistical Analysis

For data analysis, we used Stata 15.0 (Stata Corp., College Station, TX, USA). For our univariate analysis, we reported mean or relative frequencies (proportions) with their standard errors (SE). For multivariable analysis, we ran three structural equation models (SEM) [62], one model for each outcome. Specific models were fitted for fatalistic cancer beliefs, perceived risk of cancer, and cancer worries. Race was the main independent variable. Gender, age, insurance status, and having a family member with cancer were the covariates. Educational attainment and income were the mediators. To test whether educational attainment and income fully explain the effect of race on outcomes, we ran models in the pooled sample, without and with educational attainment and income as mediators. Path coefficients, SE, 95% CI, z-value, and *p*-values were reported. SEM uses maximum likelihood estimates to handle missing data [63,64]. Conventional fit statistics such as the comparative fit index (CFI), the root mean square error of approximation (RMSEA), and Chi-square to degrees of freedom ratios were used. A Chi-square to degrees of freedom ratios of less than 4.00, a CFI more than 0.90, and a RMSEA of less than of 0.06 were indicators of good fit [65,66].

We did not define our mediators and outcomes as latent factors for several reasons. First, income is one and not all of the underlying mechanisms by which education improves health and behaviors. Due to labor market discrimination, differential correlations exist between educational attainment and income across racial groups. Overall, education attainment has a stronger correlation with income in Whites, as their education is more strongly rewarded by the society by high paying jobs [67,68,69]. As our findings showed, education and income differently functioned as partial mediators of the effect of race on our outcomes. Similarly, unique patterns of determinants were found for each of our outcomes, supporting our decision not to conceptualize SES and our outcomes as latent factors. Despite not having latent factors, our decision to use SEM for data analysis was based on the following advantages of SEM compared to regression models: (1) SEM more efficiently uses data, in the presence of missing data, (2) SEM enabled us to decompose the effects of race on education, income, and also the direct effects on our outcomes, (3) the error variance of the education and income were correlated, which is a feature not available in regression analysis.

## 3. Results

### 3.1. Descriptive Statistics

Table 1 summarizes descriptive characteristics among the participants. Participants had an average age of 49 years (SE = 0.34). Almost half (52%) of the participants were females. From all participants, 87% were non-Hispanic White and 13% were non-Hispanic Black. About 92% of the participants had insurance.

### 3.2. Bivariate Correlations

Race was correlated with age, education attainment, and income. Education attainment was positively correlated with income and negatively correlated with fatalistic cancer beliefs. Income was also negatively correlated with fatalistic cancer beliefs. Cancer worries and perceived risk of cancer were positively correlated, however, Cancer worries and perceived risk of cancer were not correlated with fatalistic cancer beliefs (Table 2).

### 3.3. Fatalistic Cancer Beliefs

Model 1 was performed for cancer beliefs, which showed an acceptable fit (chi^2^ = 97.276, *p* < 0.001, CFI = 0.923, RMSEA = 0.06). According to this model, race (b = 1.68; *p* < 0.001), educational attainment (b = −0.65; *p* < 0.001), and income (b = −0.33; *p* < 0.001) were all associated with cancer beliefs. Black, low educated, and low-income individuals had worse cancer beliefs. This model showed that SES indicators only partially mediate the effect of race on poor cancer beliefs. Race was directly associated with poor cancer beliefs, on top of its indirect effects through low educational attainment and low income (Table 3, Figure 1A).

### 3.4. Perceived Risk of Cancer

Model 2 was performed for perceived risk of cancer as the outcome. This model showed an acceptable fit (chi^2^ = 95.541, *p* < 0.001, CFI = 0.914, RMSEA = 0.06). According to this model, race (b = −0.55; *p* < 0.001) and income (b = 0.07; *p* = 0.005) but not educational attainment (b = 0.02; *p* = 0.714) were associated with perceived risk of cancer, with non-Hispanic Blacks and those with low income reporting lower perceived risk of cancer. This model showed that low income only partially mediates the effect of race on perceived risk of cancer. That is, race was directly associated with perceived risk, in addition to showing an indirect effect through income levels (Table 4, Figure 1B).

### 3.5. Cancer Worries

Model 3 was performed with cancer worries as the outcome. This model showed an acceptable fit as well (chi^2^ = 94.999, *p* < 0.001, CFI = 0.917, RMSEA = 0.06). According to this model, race (b = −0.36; *p* < 0.001) but not educational attainment (b = −0.05; *p* = 0.126) or income level (b = 0.02; *p* = 0.232) was associated with cancer worries. According to this model, non-Hispanic Black individuals had lower cancer worries, net of their SES. Based on this model, SES indicators did not mediate the effect of race on cancer worries. We found that race is directly associated with cancer worries, independent of educational attainment or income level (Table 5, Figure 1C).

## 4. Discussion

In a nationally representative sample of Non-Hispanic White and Black American adults, this study found that SES does not fully explain the racial differences in fatalistic cancer beliefs, perceived risk of cancer, and cancer worries. That is, race has direct effects on cancer related cognitions, emotions, and perceptions, that go beyond its effect on SES. As a result, elimination of SES gaps would not be enough for elimination of racial gap in cancer outcomes.

Low SES individuals and Blacks are at an increased risk of cancer, compared to high SES and White individuals [20,28]. Despite their higher risk, they have less accurate cancer beliefs, lower perceived risk of cancer, and less cancer worries [56,57,70,71,72,73]. This pattern suggests that Blacks may discount their risk of cancer, possibly to minimize their cognitive dissonance, particularly because cancer results in high levels of fear in them [74,75,76,77,78]. These psychological processes may contribute to low uptake of cancer screening, possibly due to avoiding cancer anxiety and worries [74,75,76,77,78,79,80]. Blacks experience other types of adversities. For instance, while age increased Whites’ chance of having a conversation about lung cancer with their doctors, Blacks’ chance of discussing lung cancer with their doctor did not increase due to ageing, which may increase the risk of undiagnosed cancer in high risk Black individuals [58]. In another study, perceived risk of cancer was associated with higher cancer screening for Whites but not Blacks [21]. It is shown that elimination of racial disparities in cancer screening may contribute to the elimination of disparities in cancer outcome, particularly mortality [23]. This combination makes the health and well-being of Black and low SES individuals at jeopardy. At the same time, this combination imposes enormous costs to the US health care system, directly and indirectly.

This is not only paradoxical but troubling. Being at high risk of cancer, combined with fatalistic cancer beliefs, low perceived risk of cancer, low cancer worries, poor cancer knowledge, low self-efficacy regarding cancer prevention is a real public health and policy challenge [37,38,39,40,61,81,82]. This challenging reality invites policy makers, public health practitioners, and clinicians to over-invest on enhancing cancer beliefs, cognitions, and emotions of low SES and Black individuals; the groups that most need these interventions, but at the same time lack them. That means, instead of universal programs, we need interventions that disproportionately target the low SES and Black individuals, instead of universal investments.

If SES could fully explain (i.e., mediate) the effects of race on health, then reducing socioeconomic disparities between racial groups would be easier, as it would be able to fully eliminate the racial inequalities in health through equalizing access of racial groups to SES resources [11]. But the reality is that such efforts, while effective, are not enough [11,41,42]. We are not arguing that such efforts are not needed, or they are not effective in reducing the racial gaps. Instead, our argument is that these differences would not be eliminated if the only focus is SES. Still, despite equal SES, racial groups will show differential outcomes [41,42]. This is mainly because SES better serve Whites than non-Whites particularly Blacks, and high SES Blacks still have high health needs [43,44,45,83,84,85]. This disadvantage of Blacks, also known as “Minorities Diminished Returns”, suggests that we tend to over-estimate the effects of enhancing SES on racial disparities [43,44,45,84,85]. The ultimate solution to racial disparities includes policies that focus on racism and structural aspects of the society, rather than merely addressing racial gaps in access to SES resources [86,87,88,89,90,91].

Racism and discrimination are possible causes why racial minority groups have worse cancer beliefs, cognitions, and emotions, above and beyond SES. Another explanation for this phenomenon may be health literacy, and cancer literacy, in particular [33,92]. Finally, some of the racial and ethnic differences in cancer beliefs, cognitions, and emotions may be due to culture [93,94,95]. Additional research is needed to decompose the role of structural and social factors, culture, and knowledge (e.g., health literacy) in racial differences that are beyond SES differences. Stigma, mistrust, and fear should not be left behind when we address race and SES disparities in cancer emotions and cognitions [96].

Elimination of SES differences across racial groups is not enough for elimination of racial gap in health, and cancer is not an exception to this rule. The effect of race outside of SES is mainly due to racism and discrimination. Society unequally treats racial groups, based on their skin colors, and any non-White group is perceived as inferior, and is discriminated against. Discrimination is a known risk factor for poor health [97,98]. Barriers beyond SES should not be ignored as a major cause of racial disparities in cancer outcomes [99]. Mass media campaigns enhance cancer control via cancer education that target marginalized groups. Such efforts should simultaneously target racial minorities and low SES people, instead of merely focusing on either SES or race. Addressing one and ignoring the other may not be the optimal solution to the existing problems.

Cancer related cognitions, emotions, and perceptions have major implications for prevention and screening. Seeking services, as well as pro-health behaviors collectively reduce prevalence and burden of cancer [93]. Such cognitions, emotions, and perceptions are among the reasons Blacks and low SES individuals have higher cancer risk, are at risk of late diagnosis, receive late diagnoses, have lower adherence to cancer screening and treatment, and die more often from cancer [93]. According to this study, race and SES jointly cause disadvantage in cancer outcomes through their effects on cancer cognitions, emotions, and perceptions. All these processes in turn contribute to the disproportionately high risk of cancer as well as high burden of cancer in low SES and Black individuals [100].

Poor access to the health care system may partially explain poor cancer outcomes of marginalized groups including low SES and Black individuals [95]. This study checked for two indicators of access to the health care. Although we did not directly measure stigma, our SES constructs correlate with stigma. Thus, our study may have indirectly captured the confounding role of access and stigma. This argument is based on the fact that individuals who regularly use health care have lower stigma and higher trust toward the health care system and health care providers [101]. Low SES individuals and Blacks have higher stigma and lower trust to the health care system [102], which is one of the reasons they have worse cancer beliefs, cognitions, and emotions as well as cancer burden [103].

### 4.1. Study Limitations and Future Research

Current study had some limitations. First, the sample size was disproportionately lower for Blacks, which may have implications for statistical power. To solve this issue, we ran all of our models within the pooled sample, rather than running models across racial groups. Second, the study was cross-sectional in design. We cannot infer causation but association. Third, this study only included individual level factors. Fourth, this study missed some potential confounders such as history of cancer. Fifth, some of the study constructs were measured using one or only a few outcomes. There is a need to study using more sophisticated and comprehensive measures that have higher reliability and validity. There is also a need to study whether these patterns differ for age groups, and cohorts. Finally, there is a need to replicate these findings for each type of cancer, and for other race and ethnic groups. Despite these methodological and conceptual limitations, this study still makes a unique contribution to the existing literature on additive effects of race and SES, on cancer beliefs, cognitions, and emotions.

The current study was limited in how it measured the dependent variables namely cancer beliefs, cancer perceived risk, and cancer worries. Cancer beliefs were measured using the following items: (1) “There’s not much you can do to lower your chances of getting cancer”, (2) “It seems like everything causes cancer”, (3) “There are so many different recommendations about preventing cancer, it’s hard to know which ones to follow”, and (4) “When I think about cancer, I automatically think about death”. While all of these four items also reflect “fatalistic cancer beliefs”, some of these items at the same time also reflect confusion about cancer information or low perceived self-efficacy in preventing cancer (“There’s not much you can do to lower your chances of getting cancer”, “There are so many different recommendations about preventing cancer, it’s hard to know which ones to follow”. The wording of some of the items may also be problematic. For example, we do not know whether the item # 2 is taken literally or not. Particularly because of the term “seems”, this item may simply suggest that there is a barrage of information out there that is hard to interpret. Item # 3 reflects cancer misbelief but may also reflect poor self-efficacy in determining the validity of cancer information. Due to the surfeit of information from various sources that are available, it can be hard for many individuals to assess the validity of the information. These items may be confounded by a sense of frustration about own ability to determine the validity of certain claims, some of which are well known for having been reversed, even by top medical facilities. The item # 4 reflects cancer beliefs but may also be an indication of the fear associated with cancer. It may or may not literally mean that all cancer diagnoses are lethal.

### 4.2. Implications

The results reported here have major implications for research, practice, and policy making. The results advocate for looking beyond SES as a root cause of cancer disparities across racial groups in the US. Although SES is one of the major contributors of racial disparities in cancer, it is not the sole factor. Racial disparities in cancer are the results of race and SES rather than race or SES. Therefore, US policies should address social and structural processes and phenomena such as racism as well as poverty and low educational attainment. Elimination of racial disparities in cancer is not simply achievable via one line of interventions that focus on SES. Instead, multi-level solutions are needed that address race as well as SES. Policies that only focus on economic and social resources are over-simplistic and will not eliminate the sustained and pervasive disparities by race and SES [41,42].

## 5. Conclusions

To conclude, only some of the racial disparities in cancer beliefs, cognitions, and emotions are due to racial differences in SES. Policy makers, practitioners, public health experts, and researchers should consider race as well as SES as factors that jointly cause disparities in cancer outcomes. Racism, discrimination, culture, access to the health care system, and other individual and contextual factors may have a role in shaping racial disparities in cancer outcomes.

## Figures and Tables

**Figure 1 healthcare-07-00017-f001:**
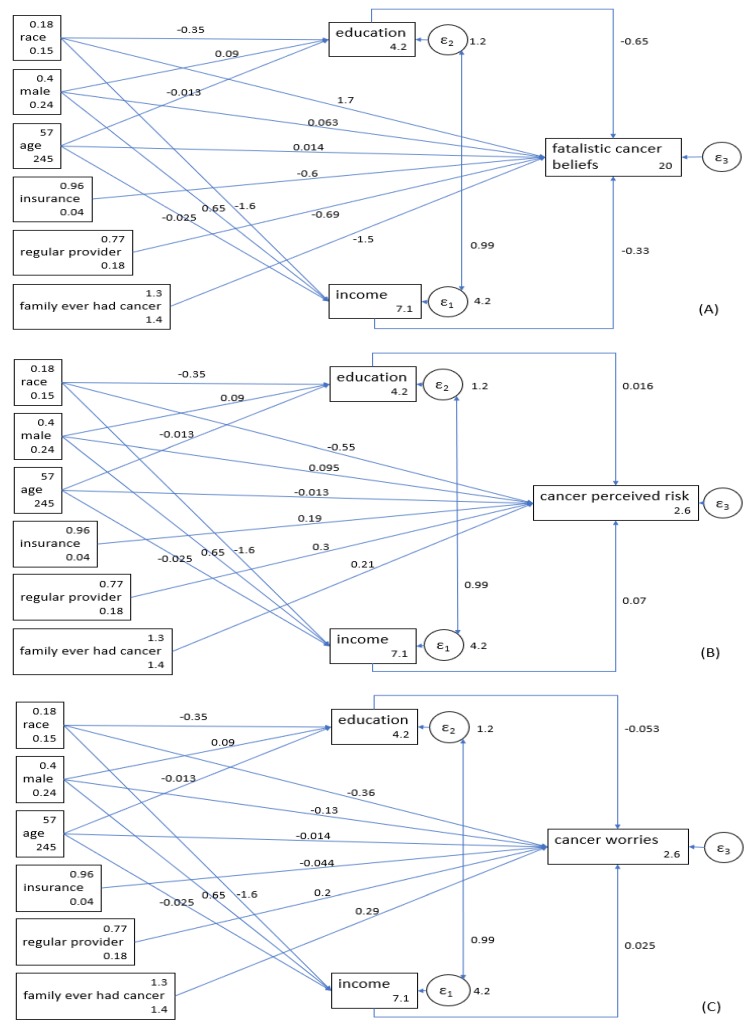
The associations between race, socioeconomic status (SES; educational attainment and income), and fatalistic cancer beliefs, perceived risk, and worries in a national sample of American adults. (**A**) Outcome: Fatalistic Cancer Beliefs. (**B**) Outcome: Cancer Perceived Risk. (**C**) Outcome: Cancer Worries.

**Table 1 healthcare-07-00017-t001:** Descriptive statistics of the participants.

**Variables**	**% (SE)**	**95% CI**
Race		
Non-Hispanic Whites	86.66 (0.01)	85.48–87.85
Non-Hispanic Blacks	13.34 (0.01)	12.15–14.52
Gender		
Male	47.89 (0.01)	46.57–49.21
Female	52.11 (0.01)	5.79–53.43
Health Insurance		
No	7.87 (0.01)	6.40–9.35
Yes	92.13 (0.01)	9.65–93.60
**Variables**	**Mean (SE)**	**95% CI**
Age (Years)	48.88 (0.34)	48.19–49.56
Income	5.60 (0.05)	5.49–5.70
Educational Attainment	3.12 (0.02)	3.08–3.16
Fatalistic Cancer Beliefs		
Perceived Risk of Cancer	2.93 (0.02)	2.83–3.03
Cancer Worries	2.54 (0.04)	2.45–2.62

Notes: Source: Health Information National Trends Survey (HINTS-5), 2017. SE: Standard Error, CI: Confidence Intervals.

**Table 2 healthcare-07-00017-t002:** Correlation between the study variables.

Variables	1	2	3	4	5	6	7	8	9
1. Race (Non-Hispanic Blacks)	1.00								
2. Gender (Male)	−0.11 *	1.00							
3. Age	−0.07 *	0.07 *	1.00						
4. Health Insurance	−0.02	−0.01	0.07 *	1.00					
5. Educational Attainment	−0.10 *	0.01	−0.15 *	0.12 *	1.00				
6. Income	−0.28 *	0.15 *	−0.14 *	0.14 *	0.46 *	1.00			
7. Fatalistic Cancer Beliefs	0.12 *	−0.01	0.04	−0.03	−0.18 *	−0.15 *	1.00		
8. Perceived Risk of Cancer	−0.14 *	0.04	−0.09 *	0.02	0.02	0.08 *	−0.01	1.00	
9. Cancer Worries	−0.06 *	−0.05	−0.11 *	−0.03	−0.02	0.00	0.05	0.29 *	1.00

* *p* < 0.05 based on Pearson correlation test.

**Table 3 healthcare-07-00017-t003:** Associations between race, socioeconomic status (SES; educational attainment and income), and fatalistic cancer beliefs in a nationally representative sample of American adults.

Variables	B(SE)	95% CI	z	*p*
Fatalistic Cancer Beliefs				
Income	−0.33 (0.09)	(−0.50, −0.15)	−3.73	<0.001
Educational Attainment	−0.65 (0.15)	(−0.94, −0.35)	−4.31	<0.001
Race (Non-Hispanic Black)	1.68 (0.40)	(0.90, 2.45)	4.23	<0.001
Gender (Male)	0.06 (0.32)	(−0.56, 0.69)	0.20	0.843
Age (Years)	0.01 (0.01)	(−0.01, 0.04)	1.29	0.196
Family Member with Cancer	−1.52 (0.12)	(−1.76, −1.28)	−12.55	<0.001
Any Health Insurance	−0.60 (0.74)	(−2.04, 0.85)	−0.81	0.417
Having a Regular Source of Health Care	−0.69 (0.36)	(−1.41, 0.02)	−1.90	0.057
Intercept	19.68 (1.06)	(17.61, 21.76)	18.58	<0.001
Educational Attainment				
Race (Non-Hispanic Black)	−0.35 (0.06)	(−0.47, −0.23)	−5.70	<0.001
Gender (Male)	0.09 (0.05)	(0.00, 0.18)	1.86	0.063
Age	−0.01 (0.00)	(−0.02, −0.01)	−8.47	<0.001
Intercept	4.15 (0.09)	(3.97, 4.33)	45.63	<0.001
Income				
Race (Non-Hispanic Black)	−1.65 (0.12)	(−1.87, −1.42)	−14.19	<0.001
Gender (Male)	0.65 (0.09)	(0.47, 0.83)	7.12	<0.001
Age (Years)	−0.03 (0.00)	(−0.03, −0.02)	−8.79	<0.001
Intercept	7.12 (0.17)	(6.78, 7.46)	4.94	<0.001

Source: The Health Information National Trends Survey (HINTS) 2017, SE: Standard Error, CI: Confidence Interval.

**Table 4 healthcare-07-00017-t004:** The associations between race, socioeconomic status (SES; educational attainment and income), and perceived risk of cancer in a nationally representative sample of American adults.

Variables	b (SE)	95% CI	z	*p*
Perceived Risk of Cancer				
Income	0.07 (0.02)	(0.02, 0.12)	2.81	0.005
Educational Attainment	0.02 (0.04)	(−0.07, 0.10)	0.37	0.714
Race (Non-Hispanic Black)	−0.55 (0.11)	(−0.77, −0.33)	−4.84	<0.001
Gender (Male)	0.10 (0.09)	(−0.08, 0.27)	1.07	0.283
Age (Years)	−0.01 (0.00)	(−0.02, −0.01)	−4.31	<0.001
Family Member with Cancer	0.21 (0.03)	(0.14, 0.28)	5.99	<0.001
Any Health Insurance	0.19 (0.21)	(−.23, 0.60)	0.88	0.380
Having a Regular Source of Health Care	0.30 (0.10)	(0.10, 0.50)	2.90	0.004
Intercept	2.63 (0.30)	(2.04, 3.21)	8.78	<0.001
Educational Attainment				
Race (Non-Hispanic Black)	−0.35 (0.06)	(−0.47, −0.23)	−5.73	<0.001
Gender (Male)	0.09 (0.05)	(0.00, 0.18)	1.87	0.062
Age (Years)	−0.01 (0.00)	(−0.02, −0.01)	−8.47	<0.001
Intercept	4.15 (0.09)	(3.97, 4.33)	45.66	<0.001
Income				
Race (Non-Hispanic Black)	−1.64 (0.12)	(−1.87, −1.41)	−14.15	<0.001
Gender (Male)	0.65 (0.09)	(0.47, 0.83)	7.07	<0.001
Age (Years)	−0.03 (0.00)	(−0.03, −0.02)	−8.81	<0.001
Intercept	7.12 (0.17)	(6.78, 7.46)	41.02	<0.001

Source: The Health Information National Trends Survey (HINTS) 2017, SE: Standard Error, CI: Confidence Interval.

**Table 5 healthcare-07-00017-t005:** The associations between race, socioeconomic status (SES; educational and income), and cancer worries in a nationally representative sample of American adults.

Variables	b (SE)	95% CI	z	*p*
Cancer Worries				
Income	0.02 (0.02)	(−0.02, 0.07)	1.20	0.232
Educational Attainment	−0.05 (0.03)	(−0.12, 0.02)	−1.53	0.126
Race (Non-Hispanic Black)	−0.36 (0.09)	(−0.54, −0.18)	−3.85	<0.001
Gender (Male)	−0.13 (0.07)	(−0.27, 0.01)	−1.83	0.068
Age (Years)	−0.01 (0.00)	(−0.02, −0.01)	−5.52	<0.001
Family Member with Cancer	0.29 (0.03)	(0.24, 0.35)	1.28	<0.001
Any Health Insurance	−0.04 (0.17)	(−0.38, 0.29)	−0.25	0.799
Having a Regular Source of Health Care	0.20 (0.08)	(0.04, 0.37)	2.42	0.016
Intercept	2.87 (0.25)	(2.38, 3.35)	11.58	<0.001
Educational Attainment				
Race (Non-Hispanic Black)	−0.35 (0.06)	(−0.47, −0.23)	−5.72	<0.001
Gender (Male)	0.09 (0.05)	(0.00, 0.19)	1.89	0.058
Age (Years)	−0.01 (0.00)	(−0.02, −0.01)	−8.51	<0.001
Intercept	4.15 (0.09)	(3.97, 4.33)	45.73	<0.001
Income				
Race (Non-Hispanic Black)	−1.64 (0.12)	(−1.87, −1.41)	−14.17	<0.001
Gender (Male)	0.65 (0.09)	(0.47, 0.83)	7.09	<0.001
Age (Years)	−0.03 (0.00)	(−0.03, −0.02)	−8.91	<0.001
Intercept	7.14 (0.17)	(6.80, 7.48)	41.17	<0.001

Source: The Health Information National Trends Survey (HINTS) 2017, SE: Standard Error, CI: Confidence Interval.

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
