# Peer review of "Combined Effects of Race and Socioeconomic Status on Cancer Beliefs, Cognitions, and Emotions"

_healthcare, 2019, doi:10.3390/healthcare7010017_

Round 1
Reviewer 1 Report
Behavior is multi-determined and depends on biologic, psychologic, sociologic, and cultural influences in a persons life. Accordingly, it is important for modern science to use new statistical methodology to try to unpack these influences to be in a better position to target large influences and target those in the hopes of producing health behavior change.
The authors have done an exemplary job of unpacking two of the factors related to African-American carcinoma beliefs and practices, race and social class.
When I worked on Culture, Race, and Ethnicity one, on the major findings of the report, there was very little research on African-American Populations which was true. As a result there is very little in the literature that is well done research on African-Americans. This is the first study I have seen that has addressed Black people's fear of cancer and unpacked it regarding social class and race.
Author Response
Behavior is multi-determined and depends on biologic, psychologic, sociologic, and cultural influences in a persons life. Accordingly, it is important for modern science to use new statistical methodology to try to unpack these influences to be in a better position to target large influences and target those in the hopes of producing health behavior change. The authors have done an exemplary job of unpacking two of the factors related to African-American carcinoma beliefs and practices, race and social class.
Response: The authors appreciate the very positive
feedback of the reviewer. Thanks for your positive evaluation of the paper.
Reviewer 2 Report
Dear Authors,
Thank you very much for the opportunity to read your paper which discusses the combined effects of race and socioeconomic status on cancer beliefs, cognitions, and emotions. While the topic discussed in the paper is important, some additional information should be added to several sections which I outlined below:
Specifically, I suggest the following:
Introduction Section:
The introduction section is very short and does not provide sufficient background to the topic. The gap in current knowledge is established, however, the authors discuss briefly existing knowledge on the topic but do not elaborate on that. I recommend discussing the current state of knowledge on the contribution of race and socioeconomic status as contributing factors to cancer outcomes. Since this study only focused on two races/ethnicities (whites and Blacks), studies that examined the association between these races and cancer-related outcomes should be discussed here to provide a reader with enough background to determine the contribution of this study to the current evidence.
Methods Section:
In this section, the authors provided a sufficient description of the study methodology including design, sampling, and variables etc. There is no info on total sample size, which should be added. Also, under independent variable subsection, it is described that race was a dichotomous variable which included only non-Hispanic Whites and non-Hispanic Blacks. Is there any reason as to why these two races/ethnicities were selected for the study? This should be clarified.
Discussion Section:
In discussing the implications of this study, the evidence from other similar studies could have been presented to support the findings and implications of this study. Additional literature should be added to strengthen the study’s implications.
Overall, I suggest considering this paper for publication after revisions are made based on the feedback provided above.
Wishing you the best of luck!
Author Response
Response: Thank you very much for the great comments. They have definitely helped the paper.
The introduction section is very short and does not provide sufficient background to the topic. The gap in current knowledge is established, however, the authors discuss briefly existing knowledge on the topic but do not elaborate on that. I recommend discussing the current state of knowledge on the contribution of race and socioeconomic status as contributing factors to cancer outcomes. Since this study only focused on two races/ethnicities (whites and Blacks), studies that examined the association between these races and cancer-related outcomes should be discussed here to provide a reader with enough background to determine the contribution of this study to the current evidence.
Response 1: We added a section (literature review) to the introduction of the paper, so our results are built on them. This now gives a context to our paper.
In methods section, the authors provided a sufficient description of the study methodology including design, sampling, and variables etc. There is no info on total sample size, which should be added. Also, under independent variable subsection, it is described that race was a dichotomous variable which included only non-Hispanic Whites and non-Hispanic Blacks. Is there any reason as to why these two races/ethnicities were selected for the study? This should be clarified.
Response 2: We added the overall sample size to our methods.
Response 3: The only two major racial groups in the public data set of the HINS study is Blacks and Whites. In addition, the authors are interested in Back -White differences (see their research background).
In discussing the implications of this study, the evidence from other similar studies could have been presented to support the findings and implications of this study. Additional literature should be added to strengthen the study’s implications.
Response 4: We added a part on the literature review to our discussion.
Overall, I suggest considering this paper for publication after revisions are made based on the feedback provided above.
Wishing you the best of luck!
Reviewer 3 Report
The study was to examine the mediating effect of SES on the relationship between race/ethnicity and three separate outcomes (fatalistic cancer beliefs, perceived risk of cancer, cancer worries). Although research question is interesting,
For the methods part, I would like to see better justification of the analysis choice. There aren't any latent variables in this measurement model, meaning no need to worry about measurement errors. I'd like to know why the authors chose SEM over path analysis, given that the model has only observed variables.
The model has education and income as separate variables. That is, education and income independently mediate the relationship between ethnicity and three outcome variables; yet, the authors interpreted as if the model has one composite measure of SES. If the authors wanted to see SES effect, I wonder why not creating SES latent variable.
Correlation matrix and CFA results should be reported (could be done in appendix). I think the authors don't have CFA because they didn't have to run, which goes back to my first question of why SEM?
All three outcomes variables seem to be correlated - why not creating one outcome variable instead of three separate models?
Author Response
The study was to examine the mediating effect of SES on the relationship between race/ethnicity and three separate outcomes (fatalistic cancer beliefs, perceived risk of cancer, cancer worries). Although research question is interesting,
Response: Thank you very much for your evaluation and very helpful comments.
For
the methods part, I would like to see better justification of the
analysis choice. There aren't any latent variables in this measurement
model, meaning no need to worry about measurement errors. I'd like to
know why the authors chose SEM over path analysis, given that the model
has only observed variables.
The model has education and income as separate variables. That is, education and income independently mediate the relationship between ethnicity and three outcome variables; yet, the authors interpreted as if the model has one composite measure of SES. If the authors wanted to see SES effect, I wonder why not creating SES latent variable.
Response: In the new version of the paper, we are using the terms education ad income more frequently than SES when it comes to our findings. We have added a section to our statistical note explaining why we did not define SES as a latent factor. In this section, we have also justified our decision to use SEM even if there is no latent variable.
Correlation matrix and CFA results should be reported (could be done in appendix). I think the authors don't have CFA because they didn't have to run, which goes back to my first question of why SEM?
Response: We have reported CFA. CFA of all modes are good.
All three outcomes variables seem to be correlated - why not creating one outcome variable instead of three separate models?
Response: We added a section to the statistical note explaining why we have not used our outcomes as a latent factor but observed variables.
Reviewer 4 Report
A very interesting study will be a great read for the readers. The study is well conducted with a good sample size. The outcomes are well presented and easy to follow.
Author Response
A very interesting study will be a great read for the readers. The study is well conducted with a good sample size. The outcomes are well presented and easy to follow.
Response: The authors appreciate the very positive feedback of the reviewer.
Round 2
Reviewer 3 Report
Thanks for your work to respond to the comments.
The authors said they have presented result of CFA, the confirmatory factor analysis, and correlation matrix of variables that were included in measurement models, I could not find the appendix or figure presenting them. It is a convention to report them along with model statistics. Please refer to the following paper:
Schreiber, J. B., Nora, A., Stage, F. K., Barlow, E. A., & King, J. (2006). Reporting structural equation modeling and confirmatory factor analysis results: A review. The Journal of educational research, 99(6), 323-338.
Also, in the event that the authors decided not to consider reviewer's comments, they should provide an explanation of that decision.
Author Response
Comments:
The authors said they have presented result of CFA, the confirmatory factor analysis, and correlation matrix of variables that were included in measurement models, I could not find the appendix or figure presenting them. It is a convention to report them along with model statistics. Please refer to the following paper:
Schreiber, J. B., Nora, A., Stage, F. K., Barlow, E. A., & King, J. (2006). Reporting structural equation modeling and confirmatory factor analysis results: A review. The Journal of educational research, 99(6), 323-338.
Also,
in the event that the authors decided not to consider reviewer's
comments, they should provide an explanation of that decision.
Response:
Thanks a lot for the comments.
Here are the responses and description of the changes. All the changes are in yellow.
Sorry, I have reported CFI and I read CFA as CFI. Now I can explain why CFA is not helpful. Please see the correlation matrix.From the 3 outcomes, 2 of them are showing weak correlations, and one of them is not even correlated with the other two. So, there is no a latent factor here. But again, I am sorry that I mis-read CFA as CFI.
I have added bivariate correlation matrix, and a brief paragraph explaining the bivariate results. They are based on Pearson correlation test.
Again thank you!
Round 3
Reviewer 3 Report
Thanks for your hard work to address the comments. I thought I commented on the figure 1, whose numbers and variable names are not really readable. Also, what do the numbers in the variable box stand for such as 57, 245 in age box? Are they necessary information? Again, tables and figures should be stand-alone. All information should be clearly presented and explained, so that readers can understand the figure without browsing around the paper to find out notations, labels, etc.
Some places have different fonts - please clean up. Eg., bivariate correlation section.
Other than that, I'm comfortable with the current form.
Author Response
It is cleaned.
fonts are fixed.
the graphs are improved (covariances are not important).
all tables are now stand alone, and added descriptions...